# Root application of Bisphenol A (BPA) and di(2-ethylhexyl) phthalate (DEHP) at environmental doses impacts tomato growth and production

**Nina Durand, Johanna Rivas**[ID]¤**, Annick Maria, Annabelle Fuentes, Emilie Crilat, David Siaussat, Arnould Savouré**[ID]**, Cécile Cabassa**[ID]*

Sorbonne Université, UPEC, CNRS, IRD, INRAE, Institute of Ecology and Environmental Science of Paris (iEES), Paris, France

¤ Current address: Univ. Bordeaux, CNRS, Bordeaux INP, EPOC, UMR 5805, F-33600 Pessac, France
* cecile.cabassa@sorbonne-universite.fr

## Abstract

The escalating decline in biodiversity in agroecosystems is a growing concern. Chemical pollutants such as plasticizers are increasingly implicated in this decline, with potential implications for both food production and quality. Among these pollutants, the endocrine-disrupting compounds (EDC) bisphenol A (BPA) and di(2-ethylhexyl) phthalate (DEHP) are particularly widespread and well documented in agricultural environments. Previous studies have reported the bioaccumulation of BPA and DEHP in crop plants, resulting in impaired growth and productivity. However, these findings were based on contaminant concentrations thousands of times exceeding those typically found in the environment. This raised critical questions about the real effects of BPA and DEHP on crop plants at environmentally relevant doses, particularly under conditions of single or combined exposure. This study investigated the effects of environmental doses of BPA and DEHP on tomato plants (*Solanum lycopersicum*), one of the most important crops in the world. Plants were exposed via root application to low and intermediate environmental concentrations (0.05 µg.L$^{-1}$ BPA and 0.5 µg.L$^{-1}$ DEHP or 50 µg.L$^{-1}$ BPA and 10 µg.L$^{-1}$ DEHP, respectively), under both mono- and co-exposure. The results revealed disruption in plant growth and productivity, which varied based on the specific EDC, concentration and exposure conditions.

## Introduction

Bisphenol A (BPA) and di(2-ethylhexyl) phthalate (DEHP) are common components of many industrial products that have been reported to have endocrine disrupting effects on various organisms. The endocrine disrupting compounds (EDC) have been reported to interfere with endogenous hormone metabolism and transport, with effects varying depending on the specific EDC and its exposure dose [1,2]. Despite

**Data availability statement:** All relevant data are within the paper and its Supporting Information files.

**Funding:** This work was supported by the French National Research Program for Environmental and Occupational Health of Anses (2020/01/143) (ANSES) [project PE-Agro- Eval 2021-2024]. The funders had no role in study design, data collection and analysis, decision to publish, or preparation of the manuscript.

**Competing interests:** The authors have declared that no competing interests exist.

regulatory restrictions, EDCs continue to be released into the environment mainly through industrial wastes and sludges used as agricultural fertilizers [3,4]. Indeed, DEHP concentrations ranging from 0.3 to 117 mg.kg$^{-1}$ of dry soil [5,6] and BPA concentrations ranging from 0.7 to 42 µg.kg$^{-1}$ of dry soil [7,8] have been detected in agricultural soils. Data from the Quali Agro platform [9] confirmed the widespread occurrence of these two EDCs, raising serious concerns about their potential impact on agricultural ecosystems, production, and consumer health. By threatening organisms at all trophic levels, these EDCs may also exacerbate biodiversity loss.

BPA, the most widely used bisphenol, is classified as a xenoestrogen [10]. Exposure to high doses of BPA, in the mg.L$^{-1}$ range, has been associated with increased risk of cancer and reproductive system disruption in animals, and adverse effects on plant growth and development [11,12]. Similarly, DEHP exposure at comparable levels has been shown to disrupt hormone receptors and signaling pathways in vertebrates and crop pests, such as *Spodoptera littoralis,* leading to developmental defects [13,14].

Although cases of bioaccumulation of BPA and DEHP in crops, such as alfalfa, lettuce and tomato are limited, high concentrations of these EDCs have been detected in plant roots and leaves at different developmental stages [15,16]. This raised concerns about the potential entry of EDCs or their metabolites into the food chain. However, most studies to date have tested excessive doses of BPA and DEHP that do not reflect realistic environmental conditions. This highlighted the need to investigate the effects of environmentally relevant concentrations of these EDCs, both individually and in combination, on crop plant development and production. Soil contamination with BPA and DEHP is mainly caused by water sources, including industrial wastewater and rainfall. Reported EDC concentrations in rainwater or recycled water ranged from 50 ng to 600 µg.L$^{-1}$ of BPA according to [16] and [17] and from 0.5 µg to 24 µg.L$^{-1}$ of DEHP according to [18]. These concentrations could be divided into three levels: Low – (0.05 µg.L$^{-1}$ BPA and 0.5 µg.L$^{-1}$ DEHP), medium (50 µg.L$^{-1}$ BPA and 10 µg.L$^{-1}$ DEHP) and High concentrations (600 µg.L$^{-1}$ BPA and 24 µg.L$^{-1}$ DEHP).

In this study, we aimed to evaluate the effects of environmentally relevant doses of BPA and DEHP on the growth, development and fruit production of tomato plants. To address this question, we exposed plants to low and intermediate environmental concentrations of these EDCs using continuous root watering of tomato plants under mono- and co-exposure conditions. Several developmental and physiological traits were measured during the vegetative stage, including germination rate, branching, leaf number, plant size, photosynthetic pigment content and oxidative stress levels. Reproductive performance was assessed by monitoring flowering, fruit set, ripening and abscission rates as well as the number and weight of fruits.

## Materials and methods

### 1. Plant material and growth conditions

Three independent batches of *Solanum lycopersicum L.* Montfavet 63−5 hybrid F1 seeds produced by the Truffaut Garden center were used across five independent

experiments. The seeds were sown in potting soil and watered with only water until the first leaf appeared, so that plants at the same developmental stage could be selected at the beginning of each experiment. Selected plants were placed in trays to allow 8 biological replicates for each treatment per experiment. Treatments were applied by root watering every two days at the bottom of the containers. To minimize environmental bias, tray positions were randomized three times per week to equalize light exposure and air flow from air conditioning. Treatment solutions consisted of either BPA at 50 µg.L$^{-1}$ or 0.05 µg.L$^{-1}$ (high or low dose respectively) or DEHP at 10 µg.L$^{-1}$ or 0.5 µg.L$^{-1}$ (high or low dose respectively) or a mixture of BPA and DEHP (co-exposure at respectively high dose, mixing the high concentrations of each EDC, or low dose, mixing the low concentrations of each EDC) using 10,000-fold concentrated EDC dissolved in ethanol. The control condition was water containing the volume of ethanol added in the treated conditions. Biochemical studies were carried out on leaves after 35 days of EDC exposure, while reproductive stages were analyzed on plants watered with all treatments until harvest.

## 2. Germination rates

Nine biological replicates of germination experiments were carried out using three seed batches, corresponding to a total of 135 seeds per treatment. For each treatment, seeds were sown on filter paper soaked with 3 mL of solution in Petri dishes and kept in the dark at 22°C for one week. Conditions, controls and concentrations were as described above. Root emergence was checked twice daily, throughout the germination period.

## 3. Measurements of growth parameters

Stem elongation was checked each week by measuring the distance from the first leaf to the shoot apex (L). Elongation was calculated as the relative increase in length compared to the initial measurement on day 0 of each EDC exposure, using the formula: (L-L0)/L0. At the same time, branch emergence and total leaf number were also recorded. Measurements were taken up to the 35$^{th}$ day of treatment with EDCs, which often corresponded to the emergence of flower spikes. At harvest, biomass was collected for biochemical analyses. Each leaf was individually weighed before freezing in liquid nitrogen. Stems of each plant were also individually weighed. As leaves 6–9 were usually very small, a pool of these leaves was made and designated "L6+" for further biochemical analyses.

## 4. Measurements of flowering and fruiting parameters

Parameters were monitored during each watering day. For the flowering process, the time of emergence of the first flower spike, the number of flowers on this spike, and the time of the emergence of the second flower spike were recorded. As the emergence characteristics were consistent between the first and second spike, the second spike was excluded to simplify the model system. Thus, all subsequent analyses of fruit parameters were performed on the first spike of each plant only. The time of fruit emergence and the color transition from orange to red were recorded to follow the fruit maturation process. Additional evaluations included fruit production, percentage of mature fruit (red) versus early abscission, carpel number, and seed viability. These assessments were performed under each treatment condition using 8 biological replicates, with a total of 16–24 tomato fruits per treatment.

## 5. Measurements of photosynthetic pigments

Non-destructive measurements of chlorophyll content in leaves were performed using an Opti-Sciences CCM-300, every week up to 35 days of EDC exposure. Only well-developed and not senescent were measured and the level of the leaves used was recorded.

For biochemical measurements on leaves harvested in liquid nitrogen, frozen material was ground to powder before use and divided into aliquots to allow for multiple technical replicates and other biochemical measurements.

Photosynthetic pigments were extracted following [19]. Approximately 30 mg of frozen leaf powder was resuspended in 1 mL of acetone (92%) and then centrifuged at 10,000 × $g$ for 10 min at 4 °C. The supernatant was collected, adjusted to 2 mL with 80% acetone and its absorbance was checked at 460, 645 and 663 nm. The contents of chlorophyll (Chl) a, Chl b, Total Chl and carotenoids (Car) were determined according to Arnon's equations [20].

## 6. Malondialdehyde (MDA) content

Malondialdehyde (MDA) content was determined in fresh frozen leaves using the thiobarbituric acid (TBA) method [21]. 30 mg of frozen leaf powder was homogenized in 1 mL of TBA (0.5%) prepared in 20% trichloroacetic acid (TCA), and the TBA-MDA complex was obtained by heating at 95°C for 30 minutes. After cooling, cell debris was removed by centrifugation at 10,000 × $g$ for 10 min at 4 °C. The absorbance of the supernatant was checked at 532nm and 600nm and the MDA content was calculated by the difference between these two absorbances and by using the specific absorbance coefficient of 155 mM$^{-1}$.cm$^{-1}$. MDA content was expressed as µmol MDA.g$^{-1}$ FW.

## 7. Statistical analyses

Statistical tests and graphs were carried out using R 4.4.1. Normal distribution and homogeneity of variances were tested using the Shapiro and Bartlett tests, respectively. When both conditions were met, Anova test followed by post-hoc Tuckey test was performed. If the conditions were not met, the non-parametric Kruskal Wallis test was used, followed by a post-hoc Dunn test. Principal Component Analysis (PCA) was performed using the package stats and plotly. A total of 11 morpho-physiological and biochemical parameters data were used to obtain principal components (PCs) for the vegetative phase. A total of 5 morpho-physiological parameters data were used to obtain PCs for the reproductive phase. Six to eight replicate data were used to generate the correlation matrix. We standardized the range of the continuous initial variables so that each one of them contributed equally to the analysis. To interpret the PCs, we calculated the percentage contribution of each variable to each PC.

## Results

### 1. Impact of BPA and DEHP on tomato seed germination and vegetative growth

To address the question of how environmental doses of BPA and DEHP affect tomato plant growth, we first investigated their impact on seed germination rates under both mono- and co-exposure conditions.

   **1a. BPA and DEHP accelerated seed germination whatever the dose.** As shown in Fig 1, 50% of tomato seeds were germinated at approximately 92 h after sowing under control conditions. In contrast, exposure to BPA and DEHP, regardless of the EDC or concentration, significantly accelerated germination, reducing the time to reach 50% of germination to 68–72 hours. Low dose of DEHP mono-exposure triggered the fastest germination rate. Both mono- and co-exposure of BPA and DEHP reduced the time to reach 50% of germination by approximately 20 h compared to the control. However, the final germination percentage remained unaffected, reaching around 95% regardless of treatment. Consequently, only tomato seedlings germinated under control conditions were selected for subsequent analyses of growth and developmental parameters.

   **1b. Environmental doses of BPA and DEHP affected differentially vegetative growth.** To address the question of the effect of EDCs on tomato vegetative growth, measurements were made on aerial parts of the plants every week until day 35, corresponding to the typical emergence of flower spikes. Throughout the experiments, the water status of the plants remained stable due to controlled watering, and no visible leaf wilting was observed until harvest. Four key growth parameters were analyzed: stem elongation, branching, leaf number and biomass. Stem elongation was measured as the height between the first leaf and the apex, while leaf development rates were also monitored. At the end of the experiment, biomass of single leaf and stems was measured. Under our conditions, no branching of the tomato plants

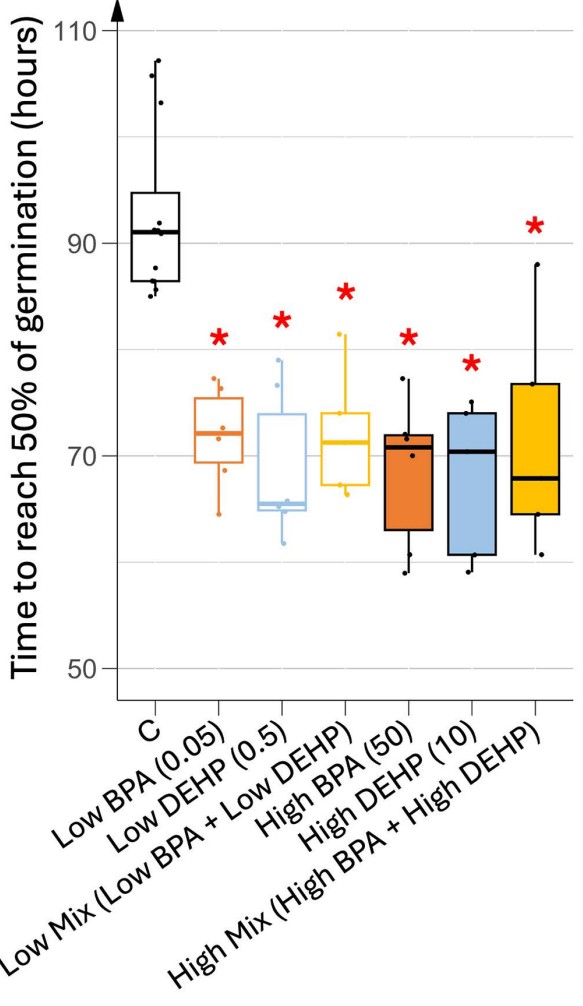

**Fig 1. Germination rates of tomato seeds exposed to various treatments.** Time to reach 50% of germination were recorded for each treatment as n = 135 using 3 different batches and 9 biological repeats. Concentrations of EDCs are indicated in brackets, units being in μg.L$^{-1}$. Stars represent significant differences between each EDC treatment against control using Kruskal-Wallis test and a post hoc of Dunn; p-value < 0.05 (*).

occurred, regardless of treatment. Both the rate of leaf formation and the final number of leaves (nine leaves before flowering) were consistent across all treatments.

While stem elongation or biomass measurements exhibited high variability between plant batches (including controls), slight but notable trends emerged.

As shown in Fig 2A, co-exposure to high concentrations of BPA and DEHP significantly reduced total aerial biomass by approximately 21% as a result of reductions in both leaves and stems. Due to high variability, other EDC treatments did not show any significant effects in total biomass (Fig 2B). However, stem biomass displayed some significant decreases when treated with low doses of BPA and high doses of DEHP (Fig 2C). Interestingly, in younger leaves (L6+), low doses in mono-exposure were associated with a slight improvement in growth, while high dose treatments had no significant effect (S1 Fig).

As shown in Fig 3, stem elongation rates during vegetative growth remained unchanged when exposed to low doses of EDC. However, higher doses, especially BPA mono-exposure significantly increased elongation rates. As stem biomass

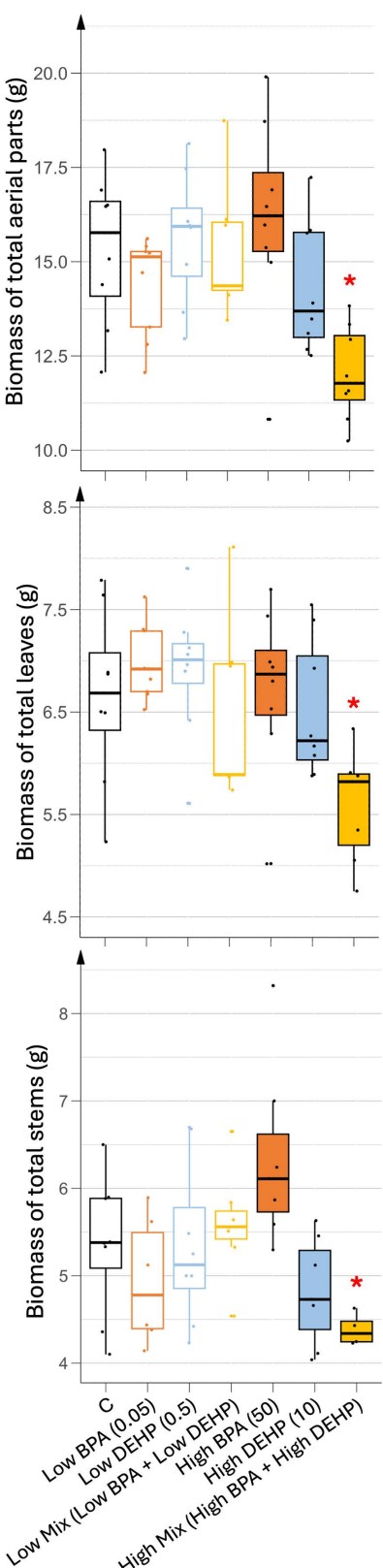

**Fig 2. Biomass of aerial parts of tomato plants exposed to various 35 days treatments.** (A) Biomass of total aerial parts, (B) Biomass of total leaves and (C) Biomass of total stems. n = 8. Concentrations of EDCs are indicated in brackets, units being in µg.L$^{-1}$. Stars represent significant differences between each EDC treatment against control using ANOVA test and a post hoc of Tukey; p-value < 0.05 (*); p-value < 0.1 (.).

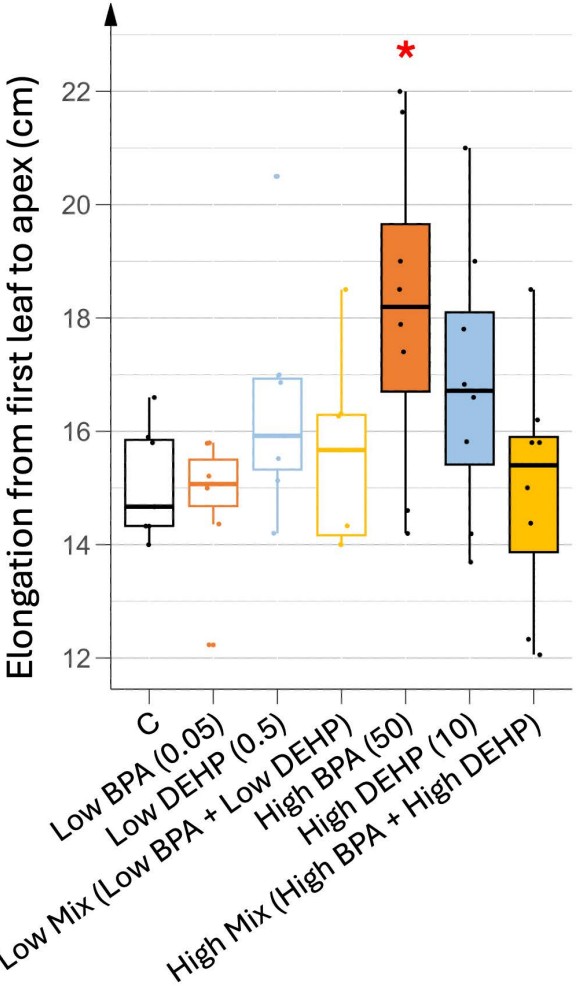

**Fig 3. Elongation of tomato stems exposed to various 35 days treatments.** Stem elongation was checked as the distance from the first leaf to the apex. n = 8. Concentrations of EDCs are indicated in brackets, units being in µg.L$^{-1}$. Stars represents significant differences between each EDC treatment against control using ANOVA test and a post hoc of Tukey; p-value < 0.05 (*).

either slightly decreased or remained unchanged in the high dose treatments, the increased elongation likely resulted in thinner stems, potentially increasing susceptibility to falling. A similar analysis was performed for the low-dose BPA treatments, where reduced stem biomass was associated with no change in elongation rate.

The high variability among control plants limited the significance of the observed differences during the vegetative phase. Nevertheless, our results indicated that high doses of BPA and DEHP negatively affect vegetative growth, while low doses may promote biomass accumulation in younger leaves. Taken together, these results suggest that BPA and DEHP exposure may disrupt key metabolic functions in tomato leaves during the vegetative phase, ultimately altering plant physiology.

## 2. Effects of BPA and DEHP on tomato physiology

In order to assess whether the differences observed during vegetative growth could be attributed to changes in physiological processes, the effects of EDCs on photosynthesis and oxidative stress were analyzed in young leaves. Photosynthetic pigment content serves as a reliable proxy for photosynthetic efficiency, as these parameters are often correlated [22]. Pigment content was measured non-destructively during the vegetative phase and destructively at the end of the experiment for leaves at different levels. Oxidative stress was also assessed at the end of the vegetative phase by measuring leaf MDA content as a proxy.

**2a. Low and high doses of EDCs had opposite physiological effects on tomato leaves.** During the vegetative phase, non-destructive chlorophyll measurements were regularly recorded on functional non-senescing leaves. However, no particular differences were observed between EDC-treated and control plants, as measurements showed high variability between conditions. Consequently, only biochemical analyses performed at the end of the experiment were included in this study.

At the end of the vegetative phase, the distribution of photosynthetic pigments (chlorophylls and carotenoids) from the oldest to the youngest leaf was similar regardless of treatment, with younger leaves showing higher pigment contents (see S2 Fig A and B). Older leaves were senescing at the time of the harvest (35 days of treatment), which explains the low pigment content found in leaves L1 to L5 and the high variability within biological replicates of these leaves. Consequently, only young developing leaves (L6 to L9, pooled as L6+) displayed reproducible replicates with statistically significant differences in photosynthetic pigment content between the different treatments (Fig 4A and B).

As shown in Fig 4, pigment contents in young leaves differed between low and high EDC treatments. Low doses of EDC significantly increased chlorophyll contents while carotenoid contents remained unchanged. In contrast, high doses of EDC resulted in a significant decrease in both pigment contents. More specifically, low doses of EDCs increased both chlorophyll a and b, whereas high doses decreased only chlorophyll a content (S3 Fig). These results were partially correlated with the vegetative growth parameters. Higher photosynthetic pigment content under low EDC dose treatments (especially BPA) was associated with increased young leaf biomass, whereas high EDC doses (especially DEHP and the co-exposure condition) correlated with reduced pigment content and aerial biomass (Fig 2). Interestingly, growth parameters (biomass and elongation) induced by high doses of BPA were inversely correlated with pigment content, highlighting the complex physiological effects of EDCs. As the vegetative growth parameters could not be fully explained by variations in the photosynthetic apparatus, the oxidative stress status in the leaves was also analyzed.

**2b. Oxidative stress increased with high single doses of BPA and DEHP.** As shown in S4 Fig, MDA accumulation increased with leaf age irrespective of treatment. Older leaves had the highest MDA content (ranging from 80 to 130 nmol.g$^{-1}$ FW), while younger leaves exhibited the lowest (ranging from 30 to 60 nmol.g$^{-1}$ FW). When analyzed individually, significant differences in MDA levels between treatments were observed in young leaves, including the early senescing L5. To facilitate comparison of oxidative stress with other biochemical parameters, data from the L6 + pool of developing leaves (L6 to L9) were analyzed.

As shown in Fig 5, a higher MDA content, marker of oxidative stress, was associated with single high doses of EDCs. This increase correlated with a decrease in pigment content at high DEHP doses and a modest reduction in growth. On the contrary, under the high co-exposure condition, MDA levels were similar to control levels, although this treatment caused the greatest reduction in aerial biomass.

## 3. EDCs disrupted fruit set and ripening

Tomato flower initiation process mainly depends on the number of leaves produced. Interestingly, each EDC treatment induced flower initiation up to two days earlier than the control plants, even when leaf formation rates were identical under all conditions. After 34 days of exposure, 50% of the plants in the EDC-treated conditions had initiated their first flower,

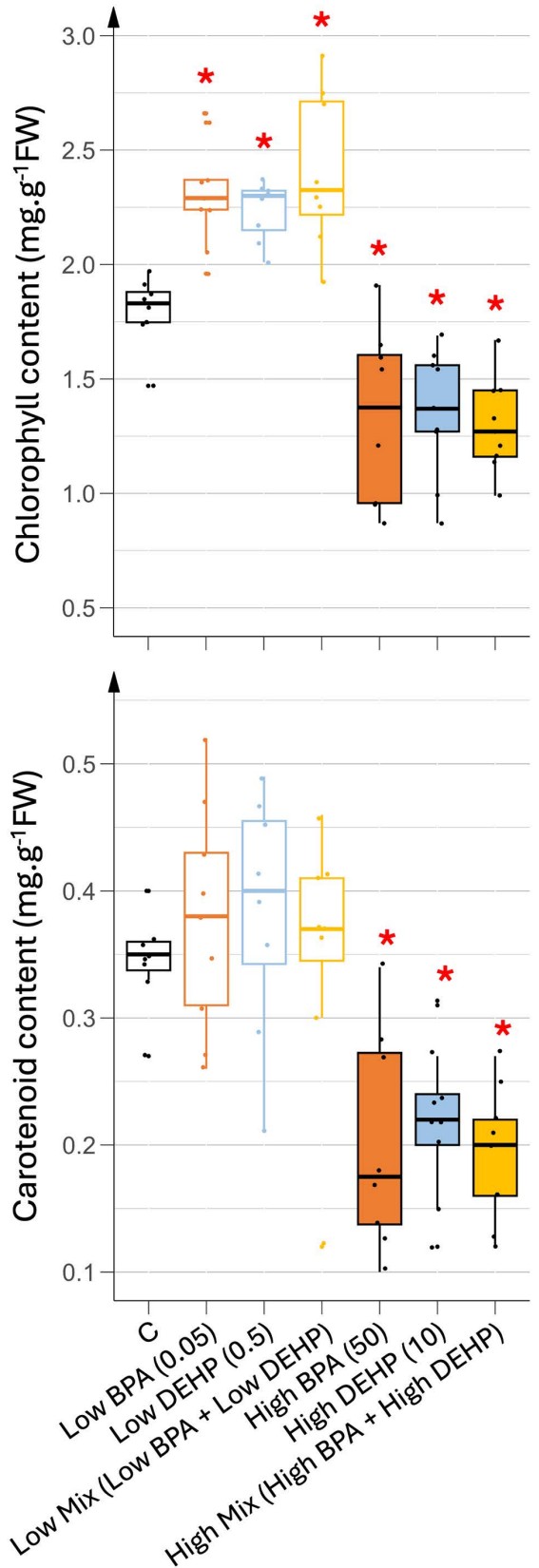

**Fig 4. Photosynthetic pigment contents in the younger leaves (L6 +) exposed to various 35 days treatments.** (A) chlorophyll (a + b) and (B) carotenoid contents. n = 8. Concentrations of EDCs are indicated in brackets, units being in µg.L⁻¹. Stars represent significant differences between each EDC treatment against control using ANOVA test and post hoc of Tukey; p-value < 0.05 (*).

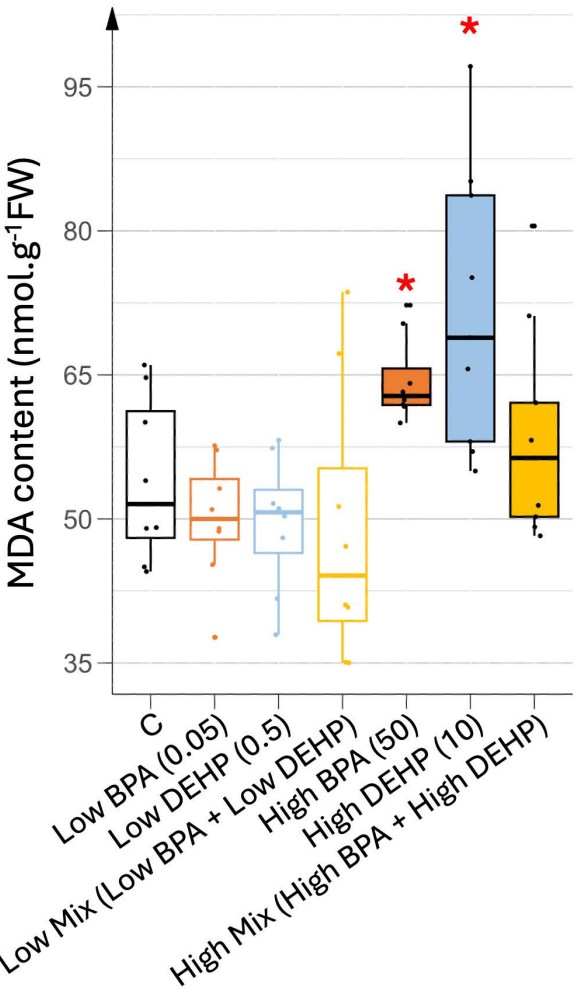

**Fig 5. MDA content in younger leaves under different EDC treatments.** MDA contents were recorded in a pool of the last younger leaves produced (leaves 6 to 9). n = 8. Concentrations of EDCs are indicated in brackets, units being in µg.L⁻¹. Stars represent significant differences between each EDC treatment against control obtained using ANOVA test and post hoc of Tukey; p-value < 0.05 (*).

in contrast to 36 days in the control plants. Similarly, the latest flower initiation occurred at day 37 in EDC-treated plants (12% of cases, except for low doses of BPA, where it occurred on day 36), whereas in controls, it occurred on day 40 (25% of cases).

In all conditions, each flower spike consistently produced six flowers, of which five or six fruits were initiated. Fruit set up occurred 6 days after flower initiation in all conditions. Control plants developed an average of two fruits per flower spike, while EDC-treated plants developed three to four fruits per spike (Fig 6). Interestingly, under co-exposure conditions, an anomalous case of single, very early fruit development, per flower spike was observed.

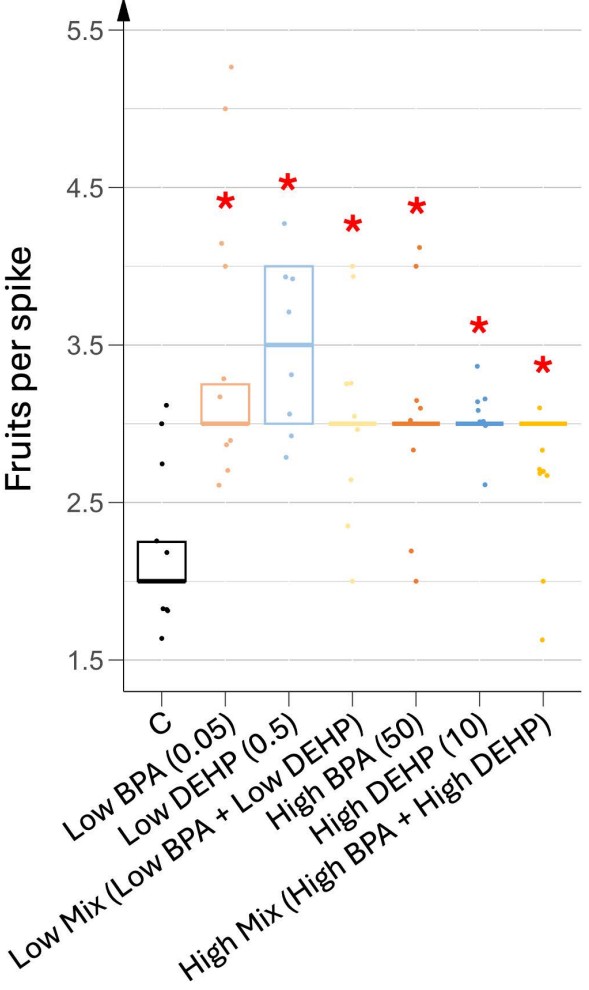

**Fig 6. Mean number of fruits per flower spike under different EDC treatments.** n = 8. Concentrations of EDCs are indicated in brackets, units being in µg.L$^{-1}$. Stars represent significant differences between each EDC treatment against control obtained using the Kruskal-Wallis test and post hoc of Dunn; p-value < 0.05 (*).

Under control conditions, all tomato fruits fell when they reached full maturity, enabling ripening to be monitored. However, under EDC treatments, only few fruits could be recorded for their time course of fruit ripening due to premature abscission. As shown in Fig 7, all EDC treatments significantly accelerated fruit ripening, especially at low doses of BPA, either mono- or co-exposed with DEHP. 75% of the control fruits ripened between 12 and 20 days, while at least 50% of each EDC treatment showed a fruit ripening time of less than 10 days. Moreover, in addition, 90% of the low doses of BPA and 100% of the fruits from plants treated with Low Mix exhibited this characteristic, with the lowest fruit ripening time falling in some cases to 5 or 6 days.

During fruit ripening, abnormal increases in the number of carpel cells were also observed when plants were treated with EDCs, with these teratological fruits representing 10% to 18% of total fruits in each EDC treatment. As shown in Fig 8, low doses of each EDC and the high dose of BPA significantly increased the mean of carpel number per fruit when compared to the control condition. All plants treated with EDCs produced teratological fruits with more than 4 carpels, the highest number being 7 in fruits from the high co-exposure treatment.

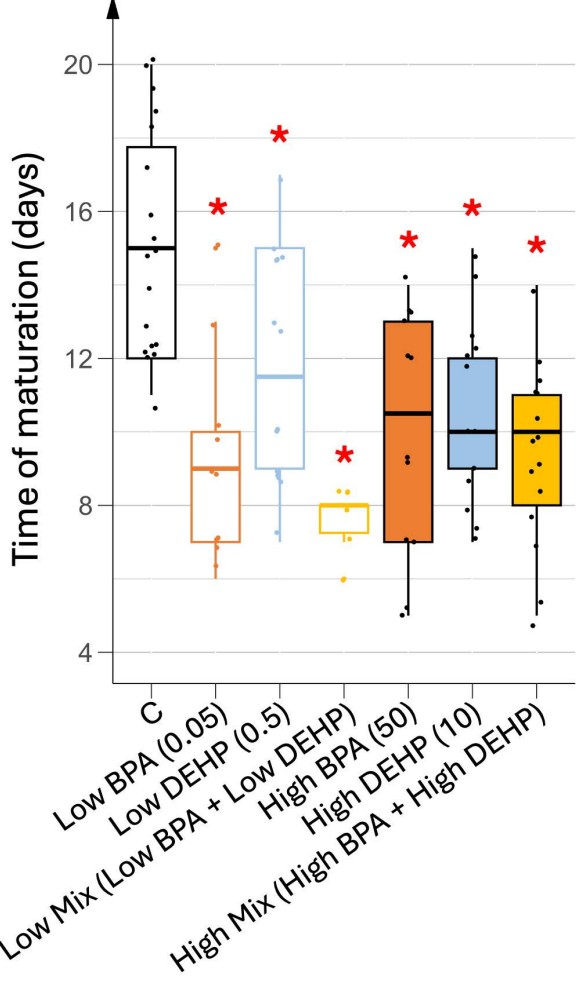

**Fig 7. Time of fruit maturation under different EDC treatments.** n = 6 to 16. Concentrations of EDCs are indicated in brackets, units being in µg.L$^{-1}$. Stars represent significant differences between each EDC treatment against control using the Kruskal-Wallis test and post-hoc of Dunn; p-value < 0.05 (*).

## 4. Effects of EDCs on early fruit abscission and production

EDCs induced premature fruit abscission, which would significantly reduce both tomato and seed production. During fruit development, early abscission of green, immature fruits was observed in all EDC treatments, regardless of doses, and in both single and combined exposures. As shown in Fig 9, fruits from control plants ripened fully before undergoing natural abscission. In contrast, early fruit abscission occurred in all treated plants, with some treatment experiencing in 100% premature abscission.

To assess seed viability, we recorded the mean number of viable seeds per flower spike, calculated as the sum of seeds from mature fruits and seeds capable of germinating on moistened paper in the dark when harvested from immature fruits. As shown in Fig 10, low doses of BPA in both mono- and co-exposures significantly reduced the number of viable seeds per spike. This reduction was attributed to early fruit abscission while mature fruits from these treated plants displayed approximately the same number of seeds as control plants, i.e., around 130 seeds per fruit. In general, the average number of seeds per mature fruit did not change regardless of EDC treatment. In plants treated with low doses of

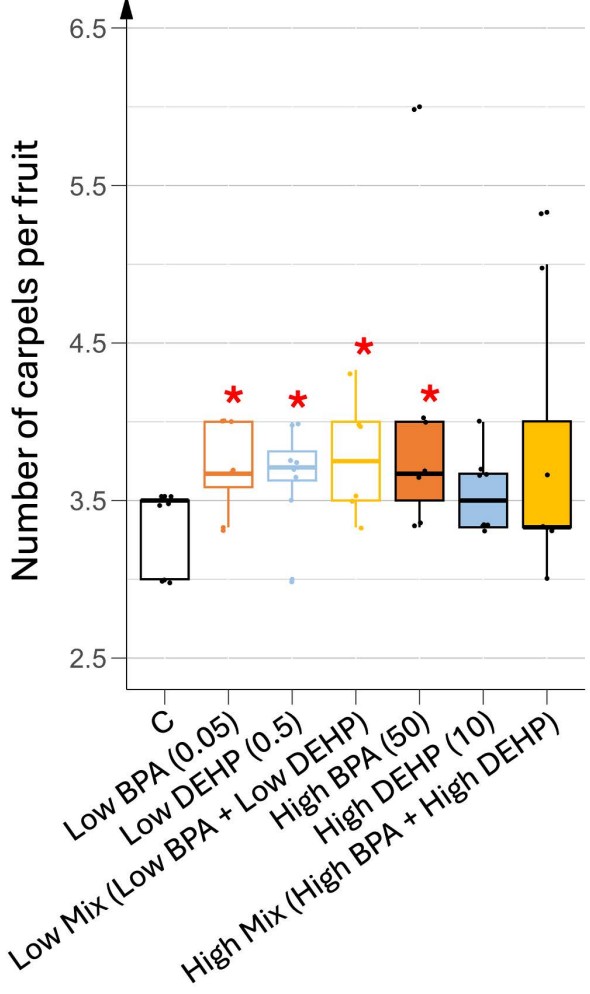

**Fig 8. Mean number of carpels per fruit under different EDC treatments.** n = 8. Concentrations of EDCs are indicated in brackets, units being in µg.L$^{-1}$. Stars represent significant differences between each EDC treatment against control using the Kruskal-Wallis test and post-hoc of Dunn; p-value < 0.05 (*).

DEHP, results were more variable due to differences in early fruit abscission, ranging from 0 to 100%. However, on closer analysis, the seed production of these DEHP-treated plants was completely different from that of the control plants. In fact, flower spikes with a high percentage of early abscission showed up to a 16% reduction in seed production, while flower spikes with no early abscission showed up to a 36% increase in seed production. In all DEHP-treated plants, regardless of the dose, approximately 20% of the fruits were completely parthenocarpic.

## 5. Principal component analyses highlighted contrasting effects of EDCs

To assess the overall effects of EDC treatments on 16 morpho-physiological and biochemical parameters, two principal component analyses (PCA) were conducted: one for the vegetative phase and another for the reproductive phase.

**5a. Effects of EDCs during the vegetative phase.** The scree plot (S5 Fig) suggested that the two PCs were parsimonious and represented the observed variance, with PC1 accounting for 60.5% (mainly representing all

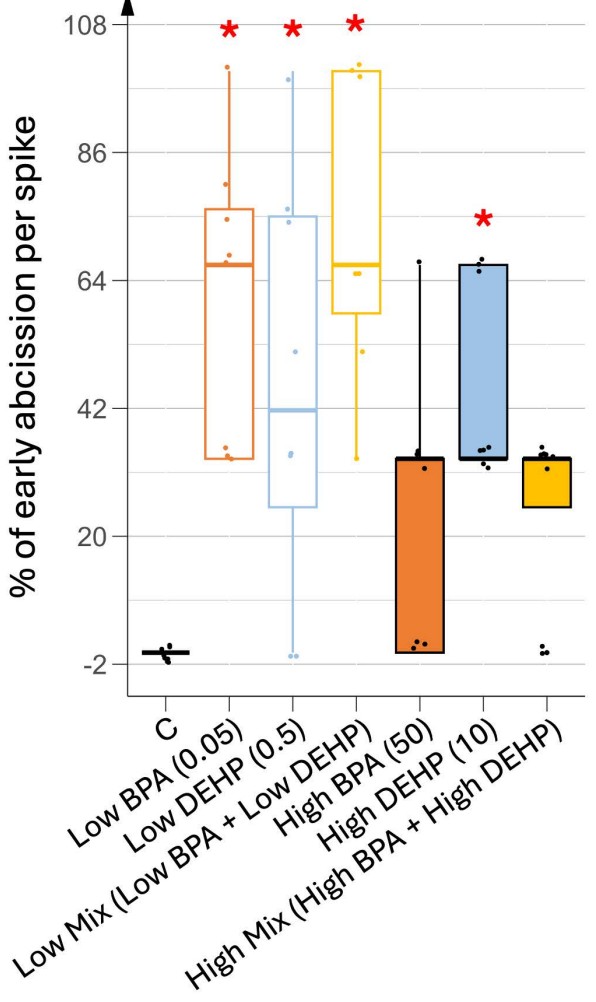

**Fig 9. Percentage of precocious fruit abscission under various EDC treatments.** n = 8. Concentrations of EDCs are indicated in brackets, units being in µg.L$^{-1}$. Stars represent significant differences between each EDC treatment against control using the Kruskal-Wallis test and post-hoc of Dunn; p-value < 0.05 (*).

photosynthetic pigments and biomasses) and PC2 for 23.3% (mainly representing MDA and stem elongation), totaling 83.8% of the variance. S6 Fig shows the correlation between the variables and the two components depending on the contribution of each variable.

The PCA (Fig 11) revealed three distinct clusters based on the concentration of the treatments applied: control, low doses and high doses of EDCs. The control cluster was positioned centrally, indicating an opposite effect of low and high doses on tomato plants, whatever the EDC or combination during the vegetative phase. In fact, high doses induced toxic effects such as a decrease in biomass and chlorophyll with higher MDA content while low doses partially stimulated the vegetative phase. The structure of the two EDC clusters was also different. Indeed, at low doses, BPA, DEHP, and their combined exposures were closely grouped, suggesting an independent effect. However, at high doses, combined exposures differed from single exposures, showing a different effect along PC1 and PC2. In fact, an additive effect was observed along PC1, where the combined high-dose treatment was positioned the farthest from the control, whereas a moderating effect was suspected along the PC2 where the combined high-dose treatment was positioned closest to the

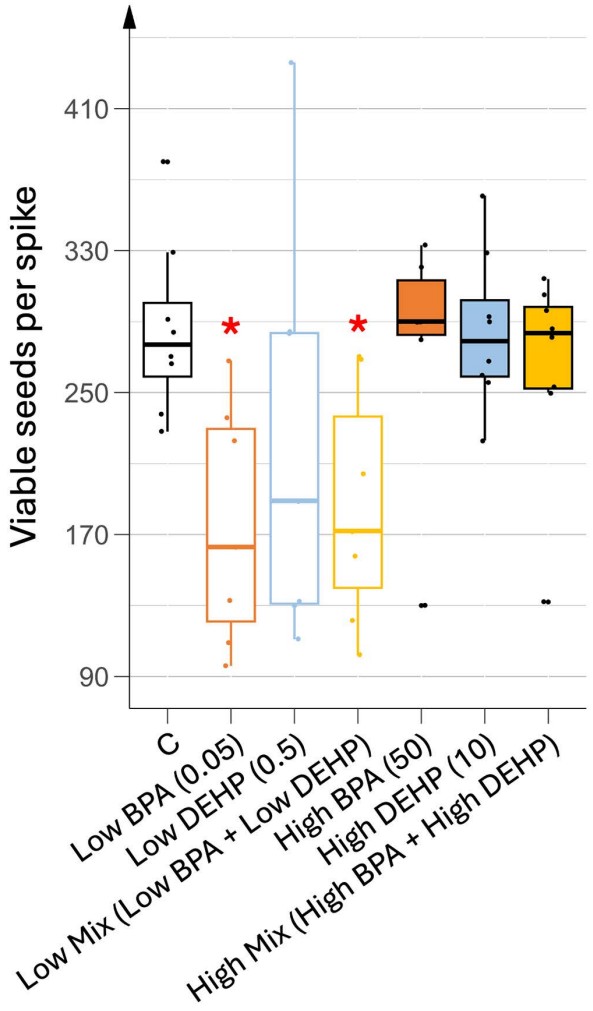

**Fig 10. Mean number of viable seeds per spike under different EDC treatments.** n = 8. Concentrations of EDCs are indicated in brackets, units being in µg.L$^{-1}$. Stars represent significant differences between each EDC treatment against control using the Kruskal-Wallis test and post-hoc of Dunn; p-value < 0.05 (*).

control. Interestingly, the high-dose cluster was further away from the control than the low-dose cluster, suggesting that high doses of EDCs had a greater effect on tomato plants during the vegetative phase.

**5b. Effects of EDCs during the reproductive phase.** For the reproductive phase, the scree plot (S7 Fig) indicated that the two PCs were optimal, with PC1 accounting for 38.1% (mainly representing early fruit abscission and the number of fruits per plant) and PC2 for 26.9% (mainly representing the time of fruit ripening and seed viability) of the total variance, giving a total of 65%. S8 Fig shows the correlation between the variables and the two components, based on their contribution.

The PCA (Fig 12) revealed two distinct groupings: the control was strongly separated from all EDC treatments. Among the treatments, low-dose BPA and mixed exposures were the furthest conditions away from the control, reflecting their significant effect in causing early premature fruit abscission but faster fruit ripening. In contrast, at high doses, combined and single exposures were indistinguishable, which differed from their effects during the vegetative phase.

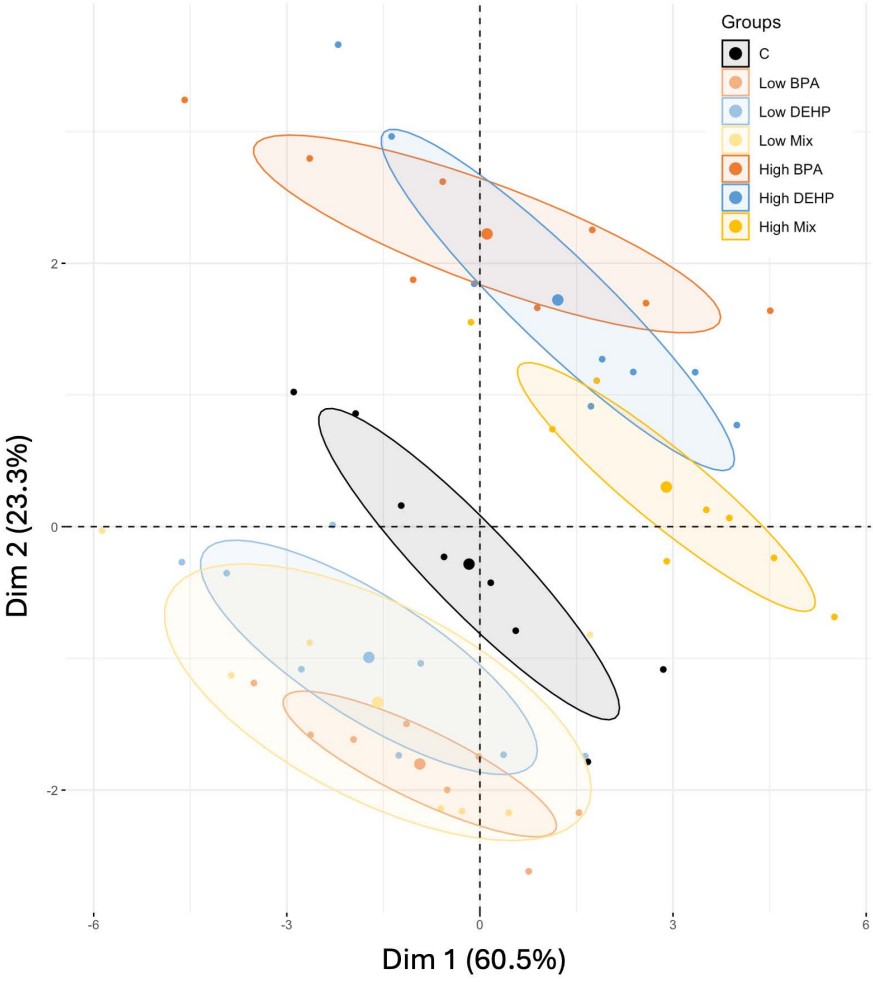

**Fig 11. PCA of the vegetative phase.** Principal Component Analysis resulting from the analysis of the behaviors of *Solanum lycopersicum* under the different EDC treatments based on 11 morpho-physiological parameters, all during the vegetative phase.

Taken together, the data recorded during the reproductive phase indicated high disturbances in all treatments, irrespective of the EDC type, dose, or combination.

## Discussion

This study demonstrated that environmental doses of plasticizers, such as BPA and DEHP, affect tomato plant germination, physiology processes, and reproductive parameters in distinct ways, depending on the type of EDCs, their dose and their combination.

### Low and high EDCs doses impaired vegetative plant growth and physiology

To isolate the effects of EDCs on growth from their impact on germination, tomato plants were treated with contaminated solutions beginning at the first leaf stage. During the vegetative phase, developmental characteristics such as branching and leaf number were not significantly affected by EDCs. However, growth parameters were markedly affected depending on the type and dose of EDCs. High doses of EDCs, especially in combined treatments drastically reduced the aerial

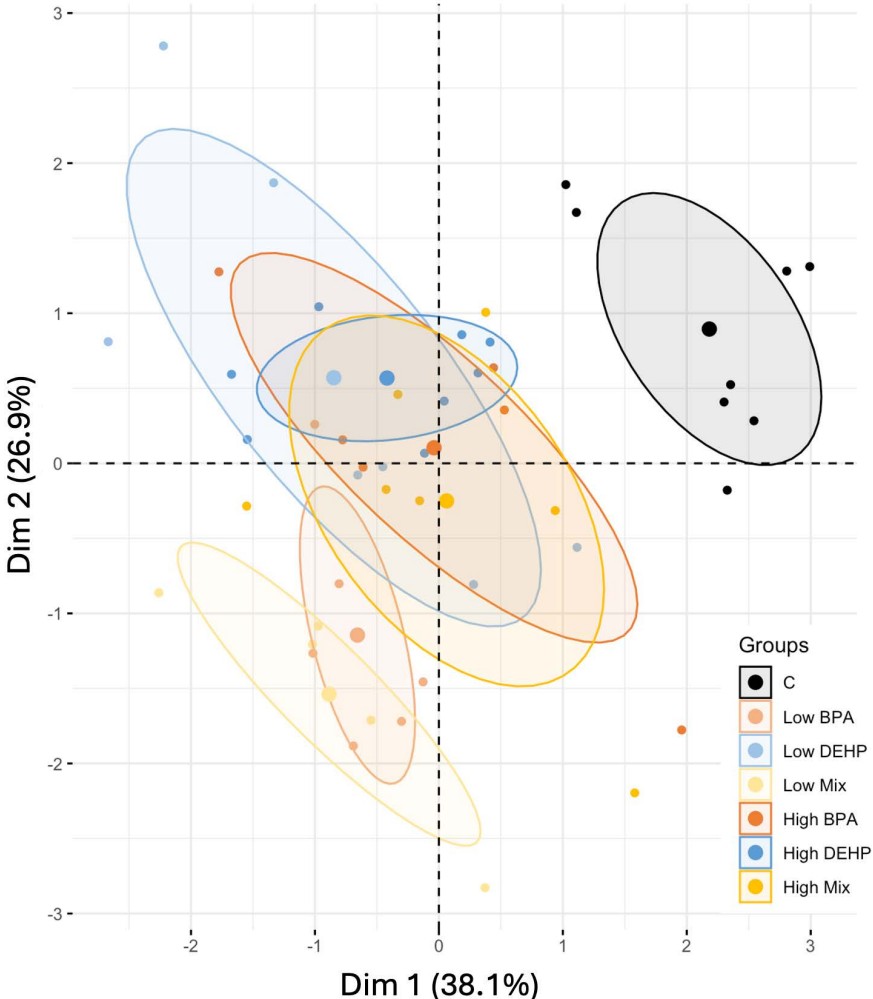

**Fig 12. PCA of the reproductive phase.** Principal Component Analysis resulting from the analysis of the behaviors of *Solanum lycopersicum* under the different EDC treatments based on 5 morpho-physiological parameters, all during the reproductive phase.

biomass of plants whereas low doses of BPA and DEHP allowed the synthesis of higher biomass than control conditions, especially with BPA. Plant elongation on the other hand was slightly increased by high single doses of EDCs, but remained unchanged by low doses of EDCs. Growth parameters are usually mainly influenced by nutrient access and uptake. As EDCs were applied at the root level of the plants, one explanation for the observed growth differences could be related to nutrient bioavailability. While the absence of earthworms or mycorrhizal fungi in the soil suggests limited biotic interactions, other microorganisms present in the rhizosphere may be affected by either BPA or DEHP. Indeed, some bacteria are known to synthesize and secrete biosurfactants (glycolipids such as rhamnolipids, lipopeptides such as surfactin, lipopolysaccharides, polymeric or particulate), siderophores or organic acids that degrade soil organic compounds to ensure their own growth (for review see [22,23]). These surface-active or extracellularly secreted molecules can also disrupt ionic interactions between ions and the soil matrix, thus modifying the bioavailability of essential nutrient (such as P, N, Ca, Mg or Fe) to plants if the microbial community involved is present in the rhizosphere (for review, see [24]). Rhizobacteria that enable enhanced plant growth or disease suppression are named Plant Growth-Promoting

Rhizobacteria (PGPR) and often function as consortia. In this context, it could be hypothesized that either BPA or DEHP may interact with the rhizospheric microbes altering bacterial metabolism or forming complexes with secreted compounds. If so, low doses of BPA and DEHP would favor the uptake of essential nutrients by tomato plants with further transport to younger leaves, while high doses would negatively affect this uptake of essential nutrients, especially under co-exposure.

The higher biomass observed in low-dose EDC treatments may correlate with an increased photosynthetic capacity allowing higher synthesis and accumulation of carbohydrates. Chlorophyll content is often used as a proxy to evaluate the photosynthetic capacity of the plants [25,26]. Here, the highest chlorophyll contents recorded in young photosynthetic tomato leaves were found in plants watered with low doses of EDCs, thus correlating with higher biomass synthesis. In the same way, these young leaves maintained their levels of carotenoids, known as photoprotective pigments as well as a control level of MDA, the peroxided lipids generated during oxidative stress and used as a proxy to evaluate the level of oxidative stress suffered by the plant [25]. Thus, low doses of EDCs may have allowed normal vegetative growth due to higher photosynthetic and maintained photoprotective capacity probably resulting in a limited oxidative stress. In contrast, all high-dose treatments reduced chlorophyll and carotenoid levels in young leaves, resulting in increased MDA levels only in mono-exposed plants. Thus, single and mixed high dose treatments may have completely different responses at the physiological and growth level. In fact, plants exposed to a mixture of high doses of EDCs exhibited a substantial reduction in biomass, but did not display elevated oxidative stress levels compared to other treatments. This discrepancy suggests that oxidative stress may not be the primary driver of growth inhibition under these conditions. One possible explanation could be a greater modification of rhizosphere activity under a mixture of high doses of EDCs only, leading to a more drastic reduction of nutrient bio-availability and/or transport inhibition in these plants but in an independent manner to any ROS signaling pathway. Another plausible explanation could be different energy allocation trade-offs between ROS defense and growth pathways in response to the different treatments used, as under co-exposition of high EDCs doses. This reallocation of energy toward oxidative stress defense could compromise photosynthetic efficiency and ultimately could result in reduced biomass accumulation. On the contrary, plants watered with a high single dose of either BPA or DEHP were not (or only slightly) altered in their vegetative growth in terms of biomass, while they accumulated a higher oxidative stress and showed a decrease in photosynthetic and photoprotective capacities, thus indicating that their failure in the defense response may have preserved enough energy to maintain biomass synthesis but not to keep the plant healthy.

Plant elongation is known to be strongly influenced by hormones such as gibberellic acid (GA), brassinosteroids, and auxins, which regulate cell elongation and organ growth. GA, synthetized in chloroplasts of young tissues, shares structural similarities, such as aromatic ring systems, with BPA and DEHP molecules, suggesting potential molecular interference. Brassinosteroids are also known to act synergistically with GA and auxin in stem elongation [27]. In our study, stem elongation was increased when plants were treated with high doses of either BPA or DEHP, possibly by disrupting GA or brassinosteroid signaling pathways. This hormonal disruption may also explain the broader effects of EDCs on other phases of the tomato plant life cycle, particularly those regulated by complex hormonal networks.

## EDCs as potential hormonal disruptors in tomato plants

BPA and DEHP are well-documented endocrine disruptors in vertebrates, known for interfering with steroid hormone pathways [28]. Their effects have also been observed in terrestrial herbivorous insects, depending on the dose [13,14]. Similar endocrine disruption processes may occur in plants, which also produce steroid hormones such as phytoecdysone, progesterone or brassinosteroids [29]. These phytohormones, as well as hormones produced by plants such as auxins, cytokinins, abscisic acid (ABA) or GA, have structural similarities with BPA, DEHP, and its derivative MEHP. This raised the hypothesis that BPA and DEHP could alter plant hormonal status by interfering with hormone metabolism, transport, or signaling.

In this study, the germination of tomato seeds was significantly accelerated by all tested EDCs, whatever the dose or combination, with 50% of germination occurring up to 20 h earlier than control seeds. In tomato, accumulation of the soluble hormone ABA in the fleshy fruits induces seed desiccation tolerance and dormancy, thus preventing premature germination [30]. In the absence of secondary dormancy, typically induced by high temperature, tomato seeds germinate after imbibition in the dark, but also in a phytochrome-dependent controlled manner [31]. In any case, a cross-talk between different hormonal signaling pathways is required to induce tomato seed germination. Indeed, seed imbibition and light induce the degradation of ABA, which is known to enhance the synthesis of germinating-promoting hormones, such as GA and progesterone [29,30]. As BPA and DEHP have structural similarities to many plant hormones, the accelerated germination may be attributed to a potential hormonal disruption leading to increased GA and/or progesterone synthesis during seed imbibition.

During the reproductive phase, EDC-treated plants exhibited earlier flowering, although leaf development was unaffected. As the onset of flowering is dependent on the number of leaves that tomato plants produce [32], our results suggested some hormonal disruption. Similarly, differences in fruit development and ripening were also observed between control and treated plants, while these processes are well known to depend on plant hormone homeostasis [33]. Indeed, before pollination, high levels of ABA and ethylene in flowers inhibit the synthesis of GA, auxins and cytokinins. After pollination and usually fertilization, ABA and ethylene are degraded, while GA biosynthesis and accumulation are induced, allowing auxin biosynthesis in the carpels, this latter hormone being required in the tomato ovary to allow fruit set up [32]. In some cases, disruption of GA and auxin signaling after pollination can induce the production of parthenocarpic fruits [34]. In this study, the number of fruits set up per spike was significantly higher in all treated plants, regardless of dose and combination, compared to control plants. Such parthenocarpic fruits were observed in about 20% of fruits from DEHP-treated plants and 10% of fruits from co-exposed plants. Similarly, low-dose BPA-treated plants displayed significantly lower viable seeds per spike than control plants. All these results may indicate a potential hormonal imbalance that would mimic higher GA and consequently increased auxin levels as if a higher number of ovaries were either fertilized or induced in a parthenocarpic process.

Fruit growth, driven by cell division and cell expansion mediated by both GA produced by the pericarp and auxins produced by the seeds appeared unaffected in terms of final shape. However, EDC-treated plants showed a significantly higher percentage of early fruit abscission rate compared to control plants. This may indicate a potential hormonal imbalance in favor of cytokinins excess over auxins in these plants, as high cytokinin-to-auxin ratios are associated with fruit abscission and increased carpel number [32]. An additional auxin deficiency may not be relevant as EDC-treated plants produced more fruits than controls. An excess of cytokinins may also explain the increased carpel number recorded in EDC-treated plants. In tomato plants, the number of carpels is usually predetermined at the flowering stage by the regulation of the floral meristem size using the CLAVATA-WUSCHEL feedback loop which hence determines the final number of carpels [35]. However, cleavage wall events that separate the carpels can occur later, during fruit growth, when the hormonal cytokinin to auxin ratio is increased. In our study, control fruits contained 3–4 carpels, already more than the 2 carpels usually found in wild tomato plants, as a result of the breeding process. EDC treatments led to abnormal carpel separation, resulting in a higher number of carpels and suggesting disruption in cytokinin biosynthesis, metabolism, or signaling.

During ripening, tomato fruits undergo a climacteric respiratory burst that triggers ethylene-regulated carotenoid biosynthesis [36]. Carotenoids as well as chlorophylls and hormones such as GA, ABA or strigolactones are terpenoids derived from isopentenyl diphosphate (IPP) and its allylic isomer dimethylallyl diphosphate (DMAPP), two isoprene isomers [37]. In tomato, carotenoids that accumulate during fruit ripening is the red pigment lycopene [38] and this accumulation is mainly regulated by ethylene. In this process, ABA accumulation precedes ethylene production and can induce the expression of genes involved in ethylene biosynthesis [39]. Ethylene accumulation or perception is required to initiate and allow the progression of fruit ripening [40] by repressing the Sl-ERF.E4 repressor of carotenoid biosynthesis. On the contrary, exogenous

application of auxins such as indole-3-acetic acid (IAA) reduces carotenoid biosynthesis in fruits, thereby delaying their ripening process [41]. Indeed, it has been shown that a decrease in IAA content together with an increase in its conjugated form IAA-Aspartate induces the repression of polar auxin transport and triggers fruit ripening [39,42]. Exogenous brassino-steroids have been also shown to induce ethylene production and fruit ripening in tomato [39]. Thus, the accelerated fruit ripening time observed in fruits from EDC-treated plants could potentially be related to both reduced IAA to IAA-Asp rate and elevated ethylene level. These effects may be due to endocrine disruption at the brassinosteroid level, further support-ing the hypothesis that BPA and DEHP may potentially act as hormone disruptors in tomato plants.

## Conclusion

This study clearly demonstrated that environmentally relevant doses of the EDCs BPA and DEHP could significantly disrupt growth and production of tomato plants, effects that were sometimes more severe than those previously observed at much higher concentrations. Key disruption included accelerated seed germination, oxidative stress in leaves and strikingly, a dras-tic premature abscission of fruits, which should directly compromise yield and crop viability. These findings raised substantial concerns for both human food security and agroecosystem stability, particularly given the widespread cultivation of tomato. Firstly, the two EDCs concentrations used induced dose-dependent and sometimes opposite responses of plants, a charac-teristic of hormonal disruption process, which underscored the complexity and unpredictability of plant responses to environ-mental doses of these EDCs. Further investigations on the hormonal pathways disrupted by these EDCs are thus needed. Secondly, beyond agricultural implications, this study also pointed out potential food safety risks for human health, both in terms of crop quality, due to physiological stress and chemical residues, and quantity, due to yield loss. Future studies should first assess potential bioaccumulation of these EDCs, their derivatives or any toxic compounds in edible plant tissues. Then, investigations on the downstream effects on organisms that consume contaminated crops, such as polyphagous pests and model mammals like mice, would be of great interest to better understand the indirect impact on the food chain and human health. These insights would not only clarify health risks but also support the development of regulatory guidelines and best practices for farmers and policymakers to mitigate EDC contamination in agricultural systems.

## Supporting information

**S1 Fig. Biomass of younger leaves (L6 +) of tomato plants exposed to various 35 days treatments.** A pool contain-ing leaves 6, 7, 8 and 9 were recorded. No significance was founded when Kruskal-Wallis test was used.
(TIF)

**S2 Fig. Photosynthetic pigment contents according to leaf level and whatever the treatment.** (A) chlorophyll a and (B) carotenoid contents in each leaf level ranging from the first leaf (1) produced to a pool of the last younger leaves (6) produced.
(TIF)

**S3 Fig. Photosynthetic pigment contents in the younger leaves (L6 +) exposed to various 35 days treatments.** (A) chlorophyll a and (B) chlorophyll b. Stars represent significant differences between each EDC treatment against control using (a) Welch test and Dunnett's test and (b) Kruskal-Wallis test and post hoc of Dunn; p-value < 0.05 (*); p-value < 0.1 (.).
(TIF)

**S4 Fig. MDA content in tomato plants according to leaf level and whatever the treatment.** MDA contents were recorded in each leaf level ranging from the first leaf produced (1) to a pool of the last younger leaves (6) produced. Each level is a mix of all treatments. No significance was founded when Kruskal-Wallis test was used.
(TIF)

**S5 Fig. Scree plot of percentage of explained variances per dimension during the vegetative phase.**
(TIF)

**S6 Fig. Correlation between the variables and the two principal component analysis of the vegetative phase.** 11 morpho-physiological parameters during the vegetative phase are represented in the two PCs ranging from the most contributing one in dark blue to the less contributing one in light blue, with values inside the table below.
(TIF)

**S7 Fig. Scree plot of percentage of explained variances per dimension during the reproductive phase.**
(TIF)

**S8 Fig. Correlation between the variables and the two principal component analysis of the reproductive phase.**
5 morpho-physiological parameters during the reproductive phase are represented in the two PCs ranging from the most contributing one in dark blue to the less contributing one in light blue, with values inside the table below.
(TIF)

## Acknowledgments

The authors are thankful to the French National Research Program for Environmental and Occupational Health of ANSES for supporting this study.

## Author contributions

**Conceptualization:** Arnould Savouré, Cécile Cabassa.

**Formal analysis:** Nina Durand.

**Funding acquisition:** David Siaussat.

**Investigation:** Nina Durand, Johanna Rivas, Annick Maria, Annabelle Fuentes, Emilie Crilat, Cécile Cabassa.

**Resources:** Johanna Rivas, Annick Maria, Annabelle Fuentes, Emilie Crilat.

**Supervision:** Arnould Savouré, Cécile Cabassa.

**Validation:** Cécile Cabassa.

**Writing – original draft:** Nina Durand, Cécile Cabassa.

**Writing – review & editing:** David Siaussat, Arnould Savouré, Cécile Cabassa.

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
