## [Decision Letter · Decision Letter 0]

16 Jun 2025

Dear Dr. Cabassa,

We look forward to receiving your revised manuscript.

Kind regards,

Rajesh Kumar Pathak, Ph.D.

Academic Editor

PLOS ONE

“This work was supported by the French National Research Program for Environmental and Occupational Health of Anses (2020/01/143) (ANSES) [project PE-Agro- Eval 2021-2024].”

“This work was supported by the French National Research Program for Environmental and Occupational Health of Anses (2020/01/143) (ANSES) [project PE-Agro- Eval 2021-2024].”

“This work was supported by the French National Research Program for Environmental and Occupational Health of Anses (2020/01/143) (ANSES) [project PE-Agro- Eval 2021-2024].”

Additional Editor Comments:

The manuscript has been reviewed and found to be of interest as it addresses important issues. However, the reviewers have raised several constructive points regarding data interpretation, methodological transparency and figure clarity. Major revisions are required including reframing mechanistic claims as hypotheses, improving figure readability and refining the dose-response analysis. These revisions will enhance the scientific rigor and overall quality of the study.

Reviewers' comments:

Reviewer's Responses to Questions

**Comments to the Author**

1. Is the manuscript technically sound, and do the data support the conclusions?

Reviewer #1: Yes

Reviewer #2: Yes

Reviewer #3: No

2. Has the statistical analysis been performed appropriately and rigorously?

Reviewer #1: No

Reviewer #2: Yes

Reviewer #3: Yes

3. Have the authors made all data underlying the findings in their manuscript fully available?

Reviewer #1: Yes

Reviewer #2: Yes

Reviewer #3: Yes

4. Is the manuscript presented in an intelligible fashion and written in standard English?

Reviewer #1: Yes

Reviewer #2: No

Reviewer #3: Yes

Reviewer #1: Reviewer Comments to the Author

Manuscript ID: PONE-D-25-27180

Title: Root application of Bisphenol A (BPA) and di(2-ethylhexyl) phthalate (DEHP) at environmental doses impacts tomato growth and production

Dear Authors,

I found the manuscript to be well-structured and focused on investigating the effects of environmentally relevant doses of BPA and DEHP on tomato plant physiology and reproduction. The study addresses an important gap in the current literature, as most prior research has concentrated on supra-environmental concentrations of endocrine-disrupting compounds (EDCs). By integrating physiological, biochemical, and reproductive parameters, the work provides a valuable perspective on the ecological relevance of such exposures.

After a thorough evaluation, I have identified several points that require revision before the manuscript can be considered suitable for publication.

Major Comments

1) Many of the observed effects (e.g., accelerated germination, early ripening, increased carpel number, parthenocarpy) are attributed to hormonal imbalances induced by BPA/DEHP. While these interpretations are plausible, no hormonal assays or gene expression analyses were conducted. The manuscript should reframe these statements as hypothesis-generating rather than confirmed mechanisms. Consider adding qualifiers like “suggest,” “may indicate,” or “potentially consistent with.”

2) High-dose co-exposure significantly reduced aerial biomass, but MDA levels remained comparable to control. This discrepancy is acknowledged but weakly explained. Consider further discussing possible alternative mechanisms (e.g., nutrient competition, transporter inhibition, or energy trade-offs).

3) Although independent biological replicates were used, please consider explicitly stating this in the Methods section and clarify whether data across seed batches were pooled, or if batch was treated as a blocking factor. To better interpret the variability observed in vegetative parameters, especially across different seed batches, it would be useful to include the coefficient of variation (CV) for key datasets. This will help readers assess whether observed differences are meaningful relative to underlying variability

4)The current figures convey the essential findings but could benefit from:

(i)In several figures (notably Figures 1–3, 6–10, and 11–12), treatment groups are labeled using abbreviated codes such as “BPA-L,” “Mix H,” or “DEHP-H,” which require the reader to refer back to the Methods or legend for interpretation. To improve clarity and accessibility—especially for non-specialist readers—I strongly recommend standardizing and expanding treatment labels directly within each figure or legend.For example, instead of “BPA-L,” use “Low BPA (50 ng/L),” or “Mix H” could be presented as “High Mix (BPA 50 µg/L + DEHP 10 µg/L).” This change will make the figures more self-explanatory and improve interpretability without relying heavily on cross-referencing. Ensuring consistency of treatment names across all figures will also enhance the manuscript's overall presentation quality.

(ii)In figures such as 1, 2, 3, and 12, where multiple treatments are compared using color-coded lines or bars, we recommend using a color-blind-friendly palette. Where appropriate, add visual cues like line styles or symbols to enhance accessibility and interpretation. Annotations or trend indicators to highlight key differences

Minor Comments

1) The manuscript exhibits inconsistencies in verb tense, particularly in transitions between the Results and Discussion sections. Scientific results should be consistently reported in past tense (e.g., “BPA significantly reduced biomass”), while present tense is acceptable when discussing general concepts or ongoing implications (e.g., “This suggests a possible hormonal effect”). Please revise the manuscript to ensure tense consistency within and across sections, especially in the Abstract, Methods, and Results.

Eg in abstract line number 35-40 “This raises critical questions about the real effects of BPA and DEHP on crop plants at environmentally relevant doses….” The correct line should be “This raised critical questions………..”

2) Correct typographical errors (e.g., “disrturbances” → “disturbances”; “breading” → “breeding”).

3) Standardize terminology: “pollutants,” “compounds,” and “EDCs” are used interchangeably—consider harmonizing for clarity.

4) Article/preposition misuse (e.g., “in favor to” → “in favor of”)

Reviewer #2: Recommendation: Minor Revision

Comments:

The topic is very interesting and the experiments were carefully performed by authors. This article demonstrates that environmental doses of both BPA and DEHP can significantly disrupt the growth and reproduction of tomato plant. This study is also really crucial for human food security and human health including various life-threatening diseases.

Thus, I recommend this article for publication in PLOS ONE after the following minor points are addressed by the authors.

Minor Revision:

1. As BPA, the most widely used bisphenol, is classified as a xenoestrogen; so, what is the dose limit of BPA to use in agriculture as high dose often associated to increase the risk of cancer?

2. Try to increase the resolution of the figure.

3. The conclusion is not impressive in its current form. Conclusion section of the manuscript requires further development to strengthen its impact and clarity.

4. It is recommended to include references to the recently published manuscripts on collagen-related genes to ensure the manuscript reflects the latest advancements in the field.

5. Authors should ensure that the manuscript is free from typos and grammatical errors.

Reviewer #3: Concentrations of the two treated substances in the plants were not analysed. This data is fundamental to explaining the toxicity of the two co-treated substances (BPA and DEHP) in the study.

Additionally, studies on the effects of BPA and DEHP on tomato growth and reproduction should reveal new information compared to previous research. For example, alongside MDA (a marker of oxidative stress), molecular mechanisms should be proposed through biomarker, genomic and metabolomic analyses. The concentration settings related to toxicity and changes in indicators caused by the simultaneous administration of BPA and DEHP must be reconsidered. This should include low concentrations of BPA and high concentrations of DEHP, as well as high concentrations of BPA and low concentrations of DEHP, rather than individual high concentrations. Experiments relating to the co-administration of high concentrations should be refined further to include dose-response data, particularly with regard to the accumulation of exposure concentrations within plants (especially partial accumulation in fruits or roots). This will validate the value of this study. (please refer : doi.org/10.1016/j.chemosphere.2020.126640, doi.org/10.1515/bmc-2022-0049, doi.org/10.1016/j.chemosphere.2024.141520)

**Do you want your identity to be public for this peer review?** For information about this choice, including consent withdrawal, please see our Privacy Policy

Reviewer #1: **Yes: ** Shivangi Chamoli

Reviewer #2: No

Reviewer #3: No

---

## [Author Response · Author response to Decision Letter 1]

10 Jul 2025

Response to reviewers PLOS ONE

We sincerely thank the Reviewers and the Editor for the time and effort dedicated to evaluating our manuscript, as well as for your constructive and insightful comments. Please note that all the points raised have been carefully considered. The corresponding changes in the revised manuscript are highlighted in yellow, and detailed responses to each comment are provided in our point-by-point reply.

Reviewer #1: Reviewer Comments to the Author

Manuscript ID: PONE-D-25-27180

Title: Root application of Bisphenol A (BPA) and di(2-ethylhexyl) phthalate (DEHP) at environmental doses impacts tomato growth and production

Dear Authors,

I found the manuscript to be well-structured and focused on investigating the effects of environmentally relevant doses of BPA and DEHP on tomato plant physiology and reproduction. The study addresses an important gap in the current literature, as most prior research has concentrated on supra-environmental concentrations of endocrine-disrupting compounds (EDCs). By integrating physiological, biochemical, and reproductive parameters, the work provides a valuable perspective on the ecological relevance of such exposures.

After a thorough evaluation, I have identified several points that require revision before the manuscript can be considered suitable for publication.

Major Comments

1) Many of the observed effects (e.g., accelerated germination, early ripening, increased carpel number, parthenocarpy) are attributed to hormonal imbalances induced by BPA/DEHP. While these interpretations are plausible, no hormonal assays or gene expression analyses were conducted. The manuscript should reframe these statements as hypothesis-generating rather than confirmed mechanisms. Consider adding qualifiers like “suggest,” “may indicate,” or “potentially consistent with.”

Our answer: We fully acknowledge that hormonal imbalance resulting from EDC exposure is a hypothesis rather than a confirmed mechanism. Accordingly, we have revised all related statements in the text to clearly reflect this as a hypothesis.

2) High-dose co-exposure significantly reduced aerial biomass, but MDA levels remained comparable to control. This discrepancy is acknowledged but weakly explained. Consider further discussing possible alternative mechanisms (e.g., nutrient competition, transporter inhibition, or energy trade-offs).

Our answer: In our previous MS version, we focused solely on energy trade-offs, stating that the energy allocated to reducing MDA levels was unavailable for plant growth. In the revised MS version, we have expanded this paragraph (line 480) to include additional potential mechanisms. Specifically, we now also consider rhizosphere activity, which was previously discussed in the section on plant biomass differences, as well as nutrient bioavailability and transport. These additions are done and highlighted in yellow in the revised text.

3) Although independent biological replicates were used, please consider explicitly stating this in the Methods section and clarify whether data across seed batches were pooled, or if batch was treated as a blocking factor. To better interpret the variability observed in vegetative parameters, especially across different seed batches, it would be useful to include the coefficient of variation (CV) for key datasets. This will help readers assess whether observed differences are meaningful relative to underlying variability

Our answer: We agree that the biological repeats of this study were not clearly explained in the Materials and Methods section. We have therefore added more details to this section in the revised version. We also provide a full explanation of these details below:

1/ Differences between seed batches could only be seen during germination experiments, with a slight modification, but the proportions remained the same between treatments and control conditions. Consequently, all the batches were pooled, so no blocking factor should be considered here.

2/ For all experiments involving plant growth, fructification and physiological measurements, variability due to seed differences was effectively minimized within and between individual biological replicates and across separate experiments, as follows:

- All seeds were germinated under control conditions. Then plantlets that had reached the same developmental stage, namely the one leaf stage (as described in the Materials and Methods section), were selected for further treatment applications with EDCs.

- Each experiment was conducted using a single seed batch, with eight plants (i.e. eight biological repeats) per treatment. All EDC treatments and control conditions were conducted simultaneously. In some experiments, fewer than eight biological repeats were used due to random plant mortality after the one-leaf stage. For every parameter measured, differences between treatments and control conditions were consistently maintained across the various experiments. However, random environmental effects within the greenhouse (e.g., positional effects) prevented us from pooling data across all experiments. Consequently, each boxplot presented in this study reflects the most representative dataset with eight biological repeats per treatment, as indicated in the figure legends. Thus, the remaining high variability observed sometimes and pointed out in the manuscript represents plant intrinsic variability regardless of seed batches.

We therefore believe that neither blocking factor nor coefficient of variation across seed batches need to be considered in our study. However, we fully agree that it was not clearly explained in our previous version of the manuscript and we have thus modified the Materials and Methods section to improve the description of our experiments.

4)The current figures convey the essential findings but could benefit from:

(i)In several figures (notably Figures 1–3, 6–10, and 11–12), treatment groups are labeled using abbreviated codes such as “BPA-L,” “Mix H,” or “DEHP-H,” which require the reader to refer back to the Methods or legend for interpretation. To improve clarity and accessibility—especially for non-specialist readers—I strongly recommend standardizing and expanding treatment labels directly within each figure or legend.For example, instead of “BPA-L,” use “Low BPA (50 ng/L),” or “Mix H” could be presented as “High Mix (BPA 50 µg/L + DEHP 10 µg/L).” This change will make the figures more self-explanatory and improve interpretability without relying heavily on cross-referencing. Ensuring consistency of treatment names across all figures will also enhance the manuscript's overall presentation quality.

(ii)In figures such as 1, 2, 3, and 12, where multiple treatments are compared using color-coded lines or bars, we recommend using a color-blind-friendly palette. Where appropriate, add visual cues like line styles or symbols to enhance accessibility and interpretation. Annotations or trend indicators to highlight key differences.

Our answer: All figures were modified accordingly.

To improve clarity further, details of the co-exposure solutions were added to the Materials and Methods section.

Minor comments:

Our answer: We thank the reviewer for taking the time to point out some writing errors, which we have now corrected.

1) The manuscript exhibits inconsistencies in verb tense, particularly in transitions between the Results and Discussion sections. Scientific results should be consistently reported in past tense (e.g., “BPA significantly reduced biomass”), while present tense is acceptable when discussing general concepts or ongoing implications (e.g., “This suggests a possible hormonal effect”). Please revise the manuscript to ensure tense consistency within and across sections, especially in the Abstract, Methods, and Results. Eg in abstract line number 35-40 “This raises critical questions about the real effects of BPA and DEHP on crop plants at environmentally relevant doses….” The correct line should be “This raised critical questions………..”

Our answer: inconsistencies in verb tense have now been corrected in all sections of the manuscript’s new version.

2) Correct typographical errors (e.g., “disrturbances” → “disturbances”; “breading” → “breeding”).

Our answer: Typographical errors have been corrected in the new version.

3) Standardize terminology: “pollutants,” “compounds,” and “EDCs” are used interchangeably—consider harmonizing for clarity.

Our answer: We chose “EDCs” and the term “EDCs” was thus introduced throughout the manuscript instead of “pollutants” or “compounds”.

4) Article/preposition misuse (e.g., “in favor to” → “in favor of”)

Our answer: The proposition “in favor to“ was changed into “in favor of”

Reviewer #2: Recommendation: Minor Revision

Comments:

The topic is very interesting and the experiments were carefully performed by authors. This article demonstrates that environmental doses of both BPA and DEHP can significantly disrupt the growth and reproduction of tomato plant. This study is also really crucial for human food security and human health including various life-threatening diseases.

Thus, I recommend this article for publication in PLOS ONE after the following minor points are addressed by the authors.

Minor Revision:

1) As BPA, the most widely used bisphenol, is classified as a xenoestrogen; so, what is the dose limit of BPA to use in agriculture as high dose often associated to increase the risk of cancer?

Our answer: Current European regulations on BPA, as outlined by the European Food Safety Authority (EFSA), focus primarily on limiting direct human exposure through food contact materials. However, there are currently no established guidelines or restrictions on the use of BPA in agriculture. Despite this regulatory gap, BPA concentrations ranging from 0.7 to 42 µg/Kg of dry soil are still measured in agricultural soils, primarily due to irrigation with contaminated water containing BPA levels between 50 ng to 600 µg/L. In the absence of specific limits for BPA application in agricultural practices, the aim of this study is to highlight the potential risks associated with repeated root exposure to these concentrations. Such exposure has been found to negatively affect crop production, posing a risk not only to food quality and yield, but also a potential threat to food security.

However, the impact for human health will be addressed in a subsequent study, focusing on organisms that consume BPA-contaminated crops, such as polyphagous agricultural pests and mice. This will help us to better understand the indirect risks associated with BPA in the food chain.

2) Try to increase the resolution of the figures

Our answer: All figures were modified accordingly

3) The conclusion is not impressive in its current form. Conclusion section of the manuscript requires further development to strengthen its impact and clarity.

Our answer: We fully agree that our conclusion was not enough impactful in its previous version. Conclusion was thus completely rewritten in the revised version of the manuscript.

4/ It is recommended to include references to the recently published manuscripts on collagen-related genes to ensure the manuscript reflects the latest advancements in the field.

Our answer: We agree that the overexpression of collagen-related genes in response to EDCs (and BPA in particular), which enhances the risk of breast cancer, is of great interest, as it is one of the most significant threats to human health when exposed to EDCs. However, we also believe that this is covered by our statement “risk of cancer” pointed out in line 61 of the introduction section. As the main aim of our study was to determine whether environmental doses of BPA and DEHP impact the growth and production of tomato plants, we believe that the latest advancements in this specific field of human health would be more meaningful in a study going deeper in analyzing the downstream effects on organisms that consume contaminated plants with environmental doses of BPA and DEHP, such as polyphagous pests and model mammals like mice.

5) Authors should ensure that the manuscript is free from typos and grammatical errors.

Our answer: We thank the reviewer for taking the time to point out some typographical and grammatical errors, which we have now corrected.

Reviewer #3:

Concentrations of the two treated substances in the plants were not analysed. This data is fundamental to explaining the toxicity of the two co-treated substances (BPA and DEHP) in the study.

Our answer: The aim of this study was to assess the impact of environmentally relevant doses of BPA and DEHP, which are commonly detected in agricultural practices, have a measurable impact on tomato plant growth and productivity. Although determining the extent of bioaccumulation in plant tissues would be valuable for evaluating potential risks to human health, this was not essential for achieving the current objective, which focused primarily on identifying adverse effects on crop performance.

Nevertheless, bioaccumulation data were collected and will be analyzed in a forthcoming study. This research will investigate the health effects on organisms that consume tomato plants exposed to these contaminants, including a polyphagous crop pest and mice. The aim is to gain a better understanding of the broader implications for food safety and ecosystem health.

Additionally, studies on the effects of BPA and DEHP on tomato growth and reproduction should reveal new information compared to previous research. For example, alongside MDA (a marker of oxidative stress), molecular mechanisms should be proposed through biomarker, genomic and metabolomic analyses.

Our answer: We fully acknowledge the scientific value of further molecular analyses, such as biomarker profiling, genomics, and metabolomics, in elucidating the mechanisms underlying the physiological changes observed following exposure to BPA and DEHP. These approaches would indeed provide deeper insights into the pathways affected by these compounds. However, we believe that such detailed molecular studies fall beyond the scope of the current work, which primarily focuses on the physiological and agronomic effects of environmentally relevant doses of these EDCs on tomato plants. We intend to address the molecular mechanisms in a subsequent, dedicated study that will build upon the present findings.

3/ The concentration settings related to toxicity and changes in indicators caused by the simultaneous administration of BPA and DEHP must be reconsidered. This should include low concentrations of BPA and high concentrations of DEHP, as well as high concentrations of BPA and low concentrations of DEHP, rather than individual high concentrations. Experiments relating to the co-administration of high concentrations should be refined further to include dose-response data, particularly with regard to the accumulation of exposure concentrations within plants (especially partial accumulation in fruits or roots). This will validate the value of this study. (please refer : doi.org/10.1016/j.chemosphere.2020.126640, doi.org/10.1515/bmc-2022-0049, doi.org/10.1016/j.chemosphere.2024.141520)

Our answer: We appreciate the suggestion to explore a broader range of co-exposure combinations. For this study, however, we opted to examine individual exposures and a representative combined treatment in order to reflect realistic environmental scenarios. Importantly, early fruit abscission, a key and severe outcome, was consistently observed across all tested conditions, including individual exposures. This indicates that BPA and DEHP are both capable of inducing significant adverse effects when applied individually.

While we agree that additional combinations (e.g., low BPA + high DEHP or high BPA + low DEHP) could provide further insights into potential synergistic or antagonistic interactions, we believe that the current dataset sufficiently supports our main conclusion: environmentally relevant doses of BPA and DEHP, whether applied individually or in combination, significantly impair the physiology, growth, and reproduction of tomato plants. These effects were, in fact, more pronounced than those previously reported at much higher concent

---

## [Decision Letter · Decision Letter 1]

4 Aug 2025

Root application of Bisphenol A (BPA) and di(2-ethylhexyl) phthalate (DEHP) at environmental doses impacts tomato growth and production

PONE-D-25-27180R1

Dear Dr. Cabassa,

We’re pleased to inform you that your manuscript has been judged scientifically suitable for publication and will be formally accepted for publication once it meets all outstanding technical requirements.

Kind regards,

Rajesh Kumar Pathak, Ph.D.

Academic Editor

PLOS ONE

Additional Editor Comments (optional):

The manuscript is acceptable for publication.

Reviewers' comments:

Reviewer's Responses to Questions

**Comments to the Author**

Reviewer #1: All comments have been addressed

Reviewer #2: All comments have been addressed

2. Is the manuscript technically sound, and do the data support the conclusions?

Reviewer #1: Yes

Reviewer #2: Yes

3. Has the statistical analysis been performed appropriately and rigorously?

Reviewer #1: Yes

Reviewer #2: Yes

4. Have the authors made all data underlying the findings in their manuscript fully available?

Reviewer #1: Yes

Reviewer #2: Yes

5. Is the manuscript presented in an intelligible fashion and written in standard English?

Reviewer #1: Yes

Reviewer #2: Yes

Reviewer #1: The authors have addressed all the queries and concerns raised during the peer-review process in a satisfactory and comprehensive manner. The revisions have enhanced the clarity, scientific rigor, and overall quality of the manuscript. Therefore, I recommend that the manuscript be accepted for publication.

Reviewer #2: (No Response)

**Do you want your identity to be public for this peer review?** For information about this choice, including consent withdrawal, please see our Privacy Policy

Reviewer #1: No

Reviewer #2: **Yes: ** Dr. Soumi Biswas

---

## [Editor Report · Acceptance letter]

PONE-D-25-27180R1

PLOS ONE

Dear Dr. Cabassa,

I'm pleased to inform you that your manuscript has been deemed suitable for publication in PLOS ONE. Congratulations! Your manuscript is now being handed over to our production team.

Kind regards,

on behalf of

Dr. Rajesh Kumar Pathak

Academic Editor

PLOS ONE